# The Magic Triangle in Oral Potentially Malignant Disorders: Vitamin D, Vitamin D Receptor, and Malignancy

**DOI:** 10.3390/ijms242015058

**Published:** 2023-10-11

**Authors:** Aya Khamis, Lara Salzer, Eik Schiegnitz, Roland H. Stauber, Désirée Gül

**Affiliations:** 1Department of Otorhinolaryngology Head and Neck Surgery, Molecular and Cellular Oncology, University Medical Center, 55131 Mainz, Germany; lsalzer@students.uni-mainz.de (L.S.); rstauber@uni-mainz.de (R.H.S.); 2Department of Oral and Maxillofacial Surgery, Plastic Surgery, University Medical Center of the Johannes Gutenberg—University Mainz, 55131 Mainz, Germany; schieeik@uni-mainz.de; 3Oral Pathology Department, Faculty of Dentistry, Alexandria University, Alexandria 5372066, Egypt

**Keywords:** VitD, VitD receptor, OPMD, SCC, premalignancy, white lesions, red lesions

## Abstract

OPMDs (oral potentially malignant disorders) are a group of disorders affecting the oral mucosa that are characterized by aberrant cell proliferation and a higher risk of malignant transformation. Vitamin D (VitD) and its receptor (VDR) have been extensively studied for their potential contributions to the prevention and therapeutic management of various diseases and neoplastic conditions, including oral cancer. Observational studies suggest correlations between VitD deficiency and higher cancer risk, worse prognosis, and increased mortality rates. Interestingly, emerging data also suggest a link between VitD insufficiency and the onset or progression of OPMDs. Understanding the role of the VitD–VDR axis not only in established oral tumors but also in OPMDs might thus enable early detection and prevention of malignant transformation. With this article, we want to provide an overview of current knowledge about OPMDs and VitD and investigate their potential association and ramifications for clinical management of OPMDs.

## 1. Oral Potentially Malignant Disorders 

Oral potentially malignant disorders (OPMDs) are a group of conditions characterized by abnormal changes in the oral mucosa that have the potential to progress to oral cancer. These disorders are considered to be a premalignant stage, meaning that if left untreated or unmonitored, they may develop into oral cancer [1]. The most recent classification system for OPMDs was proposed by the World Health Organization (WHO) in 2017. This classification system categorizes OPMDs based on their clinical appearance, histological features, and risk of progression to oral cancer [2] (Figure 1). The main categories of OPMDs according to the WHO classification are oral lichen planus (OLP), leukoplakia, proliferative verrucous leukoplakia (PVL), erythroplakia (EP), erythroleukoplakia (ELP), oral submucous fibrosis (OSF), actinic keratosis (AK), palatal lesions in reverse smokers, oral lupus erythematosus (OLE), and dyskeratosis congenital (DKC) [3].

Histopathological illustration providing an overview of various OPMDs, highlighting their distinct features. The lesions depicted include oral lichen planus (OLP), leukoplakia, proliferative verrucous leukoplakia (PVL), erythroplakia (EP), erythroleukoplakia (ELP), oral submucous fibrosis (OSF), actinic keratosis (AK), palatal lesions in reverse smokers, oral lupus erythematosus (OLE), and dyskeratosis congenita (DKC). Each lesion is characterized by specific histopathological attributes, serving as critical diagnostic indicators for clinicians. Understanding these morphological variations is essential for accurate diagnosis and timely intervention to prevent malignant transformation.

### 1.1. Oral Lichen Planus (OLP)

Oral lichen planus (OLP) is clinically distinguished by bilateral white reticular patches on the buccal mucosa, tongue, lip, and gingivae. Despite being a frequent non-infectious oral cavity condition, the actual cause of OLP is still unknown. OLP should be diagnosed based on both clinical and histological features. Specific clinical and histological criteria are used to diagnose lichen planus [4]. The occurrence of symmetrical, bilateral, white lesions affecting the buccal mucosa, tongue, lip, and gingiva is one of the clinical criteria. These lesions can take the form of white papular lesions or a lace-like network of slightly elevated white lines arranged in a reticular, annular, or linear pattern [5]. Erosions and ulcerations may also be present. OLP may also present itself as desquamative gingivitis in some cases.

Histopathologically, OLP is characterized by hyperkeratosis, acanthosis (epithelium thickening), and a band-like lymphocytic infiltrate in the subepithelial layer. The basal layer may show liquefaction degeneration, where the cells appear rounded and separated from each other. There may also be evidence of vacuolar degeneration of the basal and/or suprabasal cell layers, as well as keratinocyte apoptosis known as Civatte bodies. Civatte bodies, which are degenerated epithelial cells, may also be present within the inflammatory infiltrate. The inability of epithelial cells to regenerate caused by basal cell loss may cause epithelium thinning and, in rare cases, ulceration in the atrophic variety of lichen planus [4]. A mixed inflammatory infiltrate may also be detected. These clinical and histological characteristics aid in effectively diagnosing lichen planus and distinguishing it from other comparable disorders [6]. A correct diagnosis is required to properly manage and treat the lesion (Figure 1a). 

### 1.2. Leukoplakia 

Leukoplakia refers to a white patch or plaque that cannot be scraped off or attributed to any other known cause. Leukoplakia can occur in different forms, such as homogeneous leukoplakia (uniform appearance) and non-homogeneous leukoplakia (speckled or nodular appearance; Table 1). Leukoplakia is among the most prevalent and commonly studied OPMDs seen in clinical practice and population surveys (Liu et al., 2019). 

Several definitions of leucoplakia have been previously proposed. The WHO Collaborating Centre’s definition, issued in 2007, was “A predominantly white plaque of questionable risk having excluded (other) known diseases or disorders that carry no increased risk for cancer” [2]. Several evaluation criteria should be considered for clinical diagnosis of oral leukoplakia, such as the appearance of the borders and homogeneity (see Table 1) [7]. 

The histological picture of leukoplakia may vary according to its clinical presentation. Typically, it shows hyperkeratosis, thickening of the surface layer of cells, and acanthosis, which involves thickening of the underlying epithelium. Atypia and dysplasia may also be observed, including nuclear hyperchromatism, pleomorphism, loss of cellular polarity, and increased normal and abnormal mitotic activity, as well as abnormal cellular maturation and individual cell keratinization [8,9] (Figure 1b).

### 1.3. Proliferative Verrucous Leukoplakia 

Proliferative verrucous leukoplakia (PVL) is a unique and aggressive OPMD. Variable clinical presentations and histological characteristics distinguish it, and it is associated with the highest risk of developing oral cancer compared to other OPMDs. PVL often begins as one or more leukoplakia lesions and spreads to several areas over time when isolated foci merge or adjacent foci combine [10]. Due to its aggressive nature, it requires close monitoring, early discovery, and effective management to avoid malignant transformation. 

Microscopically, PVL exhibits hyperkeratosis, acanthosis, and dysplastic changes comparable to leukoplakia. It is mainly distinguished by the verrucous appearance and papillary projections. Severe dysplasia and even squamous cell carcinoma (SCC) may also be encountered [11] (Figure 1c).

### 1.4. Erythroplakia

Erythroplakia is a separate clinical entity characterized by a primarily red patch on the oral mucosa that cannot be classified clinically or pathologically as any other disease [12]. It is less common than leukoplakia but is considered to have a higher risk of progression to oral cancer. Erythroplakia’s solitary presentation distinguishes it from other disorders that affect several areas, such as erosive lichen planus, lupus erythematosus, and erythematous candidiasis [2].

Histopathologically, erythroplakia typically shows severe dysplasia and may sometimes even show carcinoma in situ (CIS) or invasive SCC. The epithelial cells exhibit almost all dysplastic criteria, especially cellular atypia, loss of cell maturation, and invasion of the underlying connective tissue in the case of SCC [13] (Figure 1d).

### 1.5. Erythroleukoplakia 

Erythroleukoplakia is a mixed lesion with white and red areas in the same patch or plaque. It carries a higher risk of malignant transformation than leukoplakia/erythroplakia alone. Histologically, it shows mixed features of leucoplakia and erythroplakia, exhibiting hyperkeratosis, acanthosis, and dysplasia similar to leukoplakia and frequently demonstrates high dysplastic features suggesting CIS or invasive carcinoma, such as severe cellular atypia, increased mitotic activity, and individual cell keratinization, as well as loss of typical cell architecture and loss of cell polarity [12] (Figure 1e).

### 1.6. Oral Submucous Fibrosis (OSF) 

Although not strictly categorized as an OPMD, OSF is characterized by the progressive fibrosis of the oral mucosa, leading to restricted mouth opening and potential malignant transformation. It is associated with chewing betel quid, a mixture containing areca nut and other ingredients. It is a well-known potentially malignant oral condition marked by fibrosis of the oral mucosa, including the submucosa [14]. In moderate to severe cases, fibrosis can spread to the oropharynx and the upper portion of the esophagus. The accepted definition published by Kerr et al. is a chronic, insidious disease that begins with a loss of fibro-elasticity of the lamina propria. Fibrosis and epithelial atrophy develop in the lamina propria and submucosa of the oral mucosa as the disease advances [15]. 

Oral submucous fibrosis is distinguished clinically by leathery mucosa, pallor, lack of tongue papillae, petechiae, and, on rare occasions, vesicles. OSF is also characterized by forming fibrous bands in the lips, cheek mucosa, and soft palate, resulting in limited mouth opening. The pathophysiology and clinical presentation of OSF are thought to be influenced by genetic predisposition and family history. The importance of early discovery, intervention, and regular monitoring of OSF in controlling the condition and preventing malignant change cannot be overstated [14].

Histopathological features include sub-epithelial fibrosis and hyalinization (glassy appearance), characterized by the excessive deposition of collagen in the sub-epithelial connective tissue, as well as excessive thickening of connective tissue fibers. Other features, such as hyperplasia, hyperkeratosis, and even atypia, may also be seen in the epithelium. The underlying connective tissue may occasionally show chronic inflammatory cell infiltration [16] (Figure 1f). 

### 1.7. Actinic Keratosis (AK) 

Actinic keratosis (AK) or actinic cheilitis is caused by extended exposure to actinic (solar) radiation, most notably ultraviolet rays. It affects exposed regions of the face, most notably the skin and vermilion of the lower lip. The particular locations impacted are critical in clinical evaluation [17]. AK is most typically seen in middle-aged, light-skinned males, particularly those who work outdoors. The disorder appears as isolated or generalized lesions with white flaking plaques or scaly patches with red regions intermingled. Less frequently, patients may notice lip dryness [18]. 

Histologically, AK shows epithelial hyperplasia or atrophy, abnormal maturation, varied degrees of ortho-keratinization or para-keratinization, cytological atypia, individual cell keratinization, and increased mitotic activity. The underlying lamina propria frequently demonstrates basophilic collagen degradation, elastosis, vasodilation, and pseudo-acanthosis. Pseudo-acanthosis, which arises from the infiltration of abnormal keratinocytes into the underlying connective tissue, can complicate the diagnosis of AK and SCC due to the presence of these cellular clusters in successive sections of the lesions [19] (Figure 1g).

### 1.8. Nicotinic Stomatitis (Palatal Lesion in Reverse Smokers) 

Oral lesions can arise in reverse smoking, a tobacco habit in which the burning end of a cigarette or cigar is retained inside the mouth. Reverse smoking is most common in India, the Caribbean, Latin America, Sardinia, and some Pacific Islands. Except for pipe smokers, reverse smokers have up to 50% of all oral cancers discovered on the hard palate, usually unaffected by other OPMDs [20]. It manifests itself clinically as thickened white plaques on the palate, mucosal nodularity, excrescences surrounding the orifices of small palatal mucosal glands, yellowish-brown staining, and erythema. Ulceration may also be seen. The lesions might appear as red, white, or mixed red and white patches over a tobacco-stained backdrop [21]. 

Histopathologically, the epithelium typically shows hyperkeratosis and acanthosis accompanied by squamous metaplasia and hyperplasia of the salivary ducts approaching the surface. The underlying connective tissue and minor salivary glands show chronic inflammatory cell infiltration. Notably, no atypia or dysplasia are encountered in the histopathological picture; hence, the possibility of malignant transformation is minimal, which may vary according to the intensity and duration of exposure to the insult [20] (Figure 1h). 

### 1.9. Oral Lupus Erythematosus (OLE) 

Oral lupus erythematosus (OLE) is a subtype of lupus erythematosus, a chronic autoimmune illness. Oral lesions occur in around 20% of cases with systemic lupus erythematosus. Clinically, OLE is comparable to OLP. OLE generally manifests as a central circular zone of atrophic mucosa surrounded by white striae. The most commonly affected areas are the buccal mucosa, palate, and lips. Malignant changes in OLE lesions are uncommon in the oral cavity [22]. Without systemic symptoms, it may be difficult to distinguish between OLP and OLE intra-orally. Hence, malignancies originating in lupus erythematosus may be misclassified as malignancies arising in OLP. A complete clinical, histological, and systemic evaluation is required to identify and discriminate between the two disorders [23]. 

Histopathological features include interface dermatitis, characterized by sub-epithelial lymphocytic infiltration. Basal cell degeneration may also be encountered, as well as thickening of the basement membrane. The epithelium may vary in its histopathological presentation according to the clinical presentation, ranging from atrophic to hyperkeratotic changes and hyperplasia [22] (Figure 1i). 

### 1.10. Dyskeratosis Congenita (DKC)

Dyskeratosis congenital (DKC), also known as Zinsser-Cole-Engman syndrome, is a rare genetic disorder characterized by telomere disruption. It is regarded as a potentially malignant condition, with affected people having a greater incidence of oral cancer. The disorder usually appears in childhood and should be evaluated and ruled out in children who present with oral leukoplakia. Attempts have been made to uncover potential markers for future malignant alterations inside DKC patients’ oral lesions. In DKC patients, early detection and thorough monitoring of oral lesions are critical for timely intervention and management [24,25].

Histopathologically, dyskeratosis congenita is characterized by the presence of dyskeratotic cells. These cells exhibit abnormal keratinization and demonstrate premature cell death, resulting in the formation of irregularly shaped keratin-filled bodies within the epithelium [24]. Other features may include epithelial atrophy, acanthosis, inflammation, and basal cell layer vacuolization. Additionally, dysplastic changes can occur, ranging from mild to severe, which may indicate an increased risk of malignant transformation. The histopathological findings can vary depending on the disease’s stage and severity and may overlap with other OPMDs [26] (Figure 1j).

A complete overview of the clinical and histopathological characteristics of OPMDs can be found in Table 2.

## 2. Preventive Measures and Treatment Options for OPMDs

Identifying effective preventative strategies and treatment options for OPMDs is very critical. Early diagnosis and risk stratification are crucial because OPMDs can develop into malignant oral cancer if left untreated. Taking proactive actions is necessary to reduce the prevalence of ‘high-risk’ OPMDs, and the risk of their malignant transformation, by employing effective preventive strategies. First, early detection and profound (histopathological) diagnosis can dramatically improve patient outcomes and prevent the development of oral cancer [27]. Second, proper OPMD management can aid in the reduction of morbidity and fatality rates linked with oral cancer. It is recommended that healthcare practitioners effectively address risk factors and prevent the development of life-threatening oral cancer by early intervention [28]. Third, OPMD prevention/effective treatment choices can decrease discomfort, pain, and functional restrictions, enhancing the overall quality of life for those affected by these problems. Timely management can also help to minimize invasive and lengthy treatment procedures, resulting in better patient experience/satisfaction. Moreover, focusing on preventative measures and early treatment saves money by lowering the burden of oral cancer treatment on healthcare systems and people [27,28,29].

## 3. Physiological Roles of Vitamin D and Vitamin D Receptor 

In the last three decades, research on Vitamin D (VitD) has gained close attention and significance, both within and outside the scientific community. VitD was initially described as a fat-soluble vitamin, but, currently, it is classified as a circulating pre-hormone. Calcitriol (1,25-dihydroxyVitD3) is the active form of VitD and a potent ligand of the VitD receptors (VDR) [30]. VitD possesses crucial physiological functions in the human body [31]. It regulates calcium and phosphorus levels in the body, aids in the absorption of these minerals from the intestine, reduces their excretion from the kidneys, and regulates bone turnover and remodeling [32]. VitD has attracted considerable interest from researchers and medical professionals due to its extraskeletal influence affecting multiple acute and chronic conditions [33].

The primary actions of VitD are mediated by a nuclear receptor, the VDR. It is involved in many essential cell functions, such as cell signaling, proliferation, apoptosis, and cell cycle [34,35]. VDR is activated by its ligand 1,25(OH)2D3/calcitriol. After VDR is stimulated/occupied by its ligand (calcitriol), the receptor acts as a transcriptional regulator via binding to VitD response elements (VDRE), leading to target gene activation. Calcitriol binding to VDR initiates VDR-RXR dimerization. The ligand-bound VDR-RXR complex then binds to VDRE at multiple regulatory positions of the target gene [30,36]. Moreover, VDR plays a significant role in regulating innate and adaptive immune responses, which is closely related to several OPMDs [37]. VDR signaling has been shown to enhance the production and release of anti-inflammatory cytokines, thereby reducing inflammation. VDR activation can also modulate the differentiation and activity of dendritic and regulatory T-cells, thereby exerting an immunomodulatory effect on the adaptive immune system [38]. 

Vitamin D exists in two primary forms in the human body: VitD 3 (cholecalciferol) and VitD 2 (ergocalciferol). Among these, VitD 3 is considered the most important form for human health due to its greater bioavailability and superior biological activity compared to VitD 2 [39,40]. Hence, unless explicitly specified otherwise, references to VitD pertain to VitD 3. Although both forms undergo the same hydroxylation steps, VitD 2 has a relatively low influence on the total serum 25-OHD level. This may be related to its lower binding affinity to target proteins [41,42]. VitD 2 is obtained in minimal amounts in certain foods, such as wild mushrooms, beef, and dairy products [31,43]. Its role only becomes significant for subjects taking VitD 2 supplements [41,42]. The human body obtains VitD mainly from two different sources: non-dietary sources (sunlight exposure) and dietary sources. Furthermore, VitD can also be obtained from a third source, pharmacological supplementation [31] (Figure 2).

The non-dietary source is the first and most important source in which VitD is produced in the skin. VitD synthesis is a complex process mainly activated by the effect of natural sunlight, specifically ultraviolet B (UVB) radiation (wavelengths from 270 to 315 nm) or artificial UVB rays [44,45]. Skin, representing the largest organ in the human body, is the primary location for the conversion of pro-VitD into a more active form (pre-VitD) [46,47]. Detailed steps of physiological VitD synthesis are shown in Figure 2 and can be found in more specialized VitD reviews [48,49].

Besides production in the skin, VitD can also be absorbed via dietary sources. It was assumed that a sufficient supply of VitD could be obtained by consuming VitD-rich foods. However, such foods are not adequately consumed worldwide (to varying degrees) to cover the human body’s need for VitD, often resulting in hypovitaminosis [43,50]. Fatty fish (such as salmon, herring, tuna, and mackerel), eggs, beef, and dairy products are foods high in VitD [31,43]. Different approaches have been implemented to combat hypovitaminosis D. Among all, food fortification is the most commonly implemented method [31,51]. 

In addition to dietary sources and cutaneous synthesis via sunlight exposure, pharmacological supplementation is another crucial source of VitD. This method is often employed to effectively and efficiently treat and/or prevent VitD deficiency [52]. Therefore, the optimal supplementation dosage has been controversially discussed for a long time. However, it is currently widely accepted that doses of more than 150,000 IU monthly are rarely necessary unless the person has a severe deficit [53,54]. The range for the daily recommended dietary allowance of VitD is 600 to 800 IU (40 IU = 1 µg) [55]. Under some circumstances, the recommended daily VitD consumption may be increased to 2000 IU to ensure adequate VitD supply or even 4000 IU in individuals with dark complexions, obesity, or other malabsorption conditions [39,56].

The biological effects of calcitriol are not limited to the bone. Calcitriol is also bound to DBP to other VDR-positive target tissue, such as the intestine, parathyroid gland, and others. It functions either in a genomic or non-genomic fashion [48]. Clinical studies also have been studying the effect of VitD deficiency and supplementation on different acute and chronic diseases, as well as cancer mortality [57,58,59,60]. Interestingly, VitD deficiency is reported to be associated, among others, with infection severity and incidence, inflammatory disease, mental health, metabolism, aging, immunological responses, carcinogenesis, and cancer prognosis [57,61,62]. VitD regulates a range of host defense mechanisms, including autophagy, apoptosis, cell differentiation, the release of cytokines and chemokines that promote inflammation, and, ultimately, the regulation of oxidative stress [40,61,63,64].

Vitamin D deficiency has a high prevalence worldwide. In Europe, 13.0% and 40.4% of the general population exhibits VitD deficiency and insufficiency, respectively [65]. Previous studies have linked VitD deficiency to a higher risk of cancer [66], as well as inflammatory conditions such as rheumatoid arthritis, inflammatory bowel disease, and multiple sclerosis [67,68]. VitD deficiency has been studied extensively to have a better understanding of its influence on various populations. Research undertaken in several countries has revealed differing insufficiency levels in the general population. According to research, roughly 41% of adults in the United States are VitD deficient, with more significant percentages found in specific ethnic groups such as African Americans and Hispanics [69,70]. Similarly, prevalence rates in European countries range from 20% to 50% [65,71]. VitD deficiency is prevalent in older individuals due to age-related decreases in the skin’s ability to synthesize VitD and lower dietary intake. According to studies, the incidence of insufficiency in this population might be as high as 70–80% in some areas [72,73]. Research revealed that individuals are not all equally prone to VitD deficiency; specific groups are at high risk of developing VitD deficiency. Among these groups are individuals with darker skin tones, obesity, and limited sun exposure, and individuals suffering from medical conditions that affect VitD metabolism and/or absorption [74]. VitD deficiency prevalence varies according to geographic location, season, and individual population features. 

## 4. Vitamin D and Head and Neck Cancer

Throughout the previous decades, VitD deficiency has been correlated to increased all-cause mortality, acute and chronic inflammatory diseases, cancer, and several autoimmune diseases [58]. It has been demonstrated that high doses of calcitriol can reduce the cancer cell proliferation rate [75,76]. That is why clinical researchers have been widely studying VitD’s relevance in cancer development, prevention, and/or treatment. Unfortunately, the studies’ outcomes have been inconsistent, resulting in an unclear situation without any general recommendations for VitD supplementation during cancer therapy. However, many observational studies showed a positive correlation between increased VitD intake and lower cancer risk and/or improved prognosis [77,78,79]. Thus, there is still a high need for further basic and clinical research addressing questions of ‘if’, ‘how’, and ‘how much’ VitD might be supplied during cancer treatment.

In summary, calcitriol, the active form of VitD, exerts multiple effects on cancer cells by binding to the VDR. The most well-described effects of calcitriol are its anti-metastatic [80], antiproliferative, pro-differentiation [81], angiogenic, apoptotic, and pro-autophagy action [30,40,64]. The anti-proliferative activity involves cell-cycle arrest in the G0/G1 phase [82], induction of differentiation [81], and promotion of apoptosis [83]. Calcitriol achieves these effects through both genomic and non-genomic pathways [78]. Calcitriol stimulates gene expression in cell-cycle regulation, such as cyclin-dependent kinase inhibitors (CKIS) and tumor suppressor genes, resulting in G0/G1 cell-cycle arrest [82] (Lo et al., 2022). Calcitriol also enhances apoptosis by inducing the expression of pro-apoptotic proteins, such as BAX, BAK, BAD, and BIM, and suppressing the expression of pro-survival proteins, such as BCL-2 [40,83]. Interestingly, the serum calcium level regulated by calcitriol also influences ion channels responsible for the influx and efflux of drugs across the cellular membrane [84,85]. In addition, calcitriol influences the tumor microenvironment by modulating the activity of immune cells [86]. It has been shown to suppress the secretion of pro-inflammatory cytokines and chemokines and to promote the differentiation of regulatory T cells, which can eventually suppress the activity of effector T cells [87]. It also inhibits the differentiation of hematopoietic progenitor cells by down-regulating the expression of CD40, CD80, and CD86 [88].

Data regarding VitD/VDR’s role in head and neck cancer (HNC) remains limited [89]. The results of studies examining the effects of VitD levels and/or supplementation on HNC patients’ outcomes are controversial. Some showed significant results [90,91,92,93,94,95,96], and others failed to show a significant correlation [91,97,98]. A limited number of studies have investigated the impacts of VitD supplementation on outcomes for cancer patients. However, similarly, the results have also been controversial [99]. Lathers et al. showed that calcidiol intake reduced the presence of CD34+ immune suppressive cells, hence improving the patients’ immune health, eventually affecting the disease prognosis [100]. Many clinical studies agreed on the effectiveness of VitD if given at high doses in intermittent intervals. Other studies also recommended combination therapies with VitD to enhance the effect of the partner drug/drugs [30,101,102].

A notable aspect of the studies reviewed is that the majority focused on either VitD status or supplementation or VDR expression/polymorphism without examining both simultaneously in the same patient and correlating them to each other. It should be noted that VitD and VDR are closely interconnected, with VitD being required for VDR activation and VDR being necessary for VitD to exert its effects on cellular processes. Although these factors can independently influence OPMDs and SCC patient outcomes, the current literature lacks investigations of their interplay. The efficacy of VitD/calcitriol in the development and treatment of OPMDs and HNSCC remains unclear due to the limited and inconclusive data available. Further investigation is necessary to determine the optimal dosage and feasibility of incorporating VitD/calcitriol into standard treatment regimens.

## 5. Vitamin D and OPMDs

Besides the widely discussed functions of VitD/VDR in cancer, researchers have been investigating the link between VitD deficiency and the development of OPMD [103]. Indeed, studies on the incidence of VitD insufficiency in people with OPMDs have found a correlation between low VitD levels and the development of these oral lesions [103,104]. Clearly, VitD deficiency is widespread among those diagnosed with OPMDs [105]. This incidence is due to various causes, including insufficient sun exposure, inadequate VitD intake, and underlying medical problems that interfere with its metabolism [1].

One study on leukoplakia patients found that people with low VitD levels had a higher likelihood of developing dysplastic alterations in the oral mucosa, indicating a possible role for VitD deficiency in the malignant transformation of leukoplakia [106]. Another study found a link between VitD deficiency and the severity of oral lichen planus, implying that low VitD levels may contribute to the advancement of this ailment [107]. Understanding and resolving the link between VitD supply and OPMDs could have substantial implications for their prevention and clinical management [101]. While there is no direct relationship between either, the potential protective effect of VitD against the development and progression of oral cancer may be correlated to its various biological functions (Figure 3), as discussed in the following. 

First, VitD has immunomodulatory properties, influencing innate as well as adaptive immune systems. It has been proposed that maintaining optimal VitD levels may help regulate the immune system’s response to precancerous changes in the oral mucosa, potentially reducing the risk of progression to oral cancer [108,109]. VitD receptors are found in the immune system and oral mucosal cells, indicating that VitD directly influences these tissues [108]. In the context of OPMDs, optimal VitD levels may help modulate the immune response in the oral mucosa, potentially reducing the risk of progression to oral cancer. Conversely, VitD deficiency decreases immunological function and increases susceptibility to chronic inflammation, encouraging the onset or progression of OPMDs [28].

Secondly, VitD can inhibit the production of pro-inflammatory cytokines, such as interleukin-6 (IL-6) and tumor necrosis factor-alpha (TNF-alpha), while promoting the production of anti-inflammatory cytokines such as interleukin-10 (IL-10) [110,111,112,113]. This balance is essential for preventing chronic inflammation, which is associated with cancer development. VitD has anti-inflammatory properties, and, thus, by reducing inflammation, VitD may help prevent the progression of OPMDs to oral cancer [114,115,116,117].

Furthermore, VitD plays a crucial role in regulating cell proliferation, cell differentiation, and apoptosis. VitD has antiproliferative properties, limiting aberrant cell proliferation and lowering the chance of malignant transformation [1,101]. VitD can inhibit the proliferation of cancer cells, induce cell cycle arrest, and suppress the expression of genes involved in cell growth and survival pathways [118,119]. It can also modulate gene expression through epigenetic mechanisms. VitD has been shown to affect DNA methylation patterns and histone modifications, which can influence gene expression in cancer development and progression [120,121,122]. Proper cell differentiation and apoptosis are essential for maintaining tissue integrity and preventing the accumulation of abnormal cells that can lead to oral cancer [123,124,125,126]. Here, VitD promotes cell differentiation, favoring a more mature and specialized phenotype. It also induces apoptosis in cells with damaged DNA or other abnormalities. By promoting proper cell differentiation and apoptosis, VitD may help prevent the progression of OPMDs to oral cancer [40,118,119]. VitD insufficiency has also been linked to an increased risk of oral cancer, highlighting the importance of VitD in OPMDs [40]. 

## 6. Discussion and Conclusions

Pre-clinical and clinical studies can improve our understanding of how premalignant lesions proceed to oral cancer. This input can be used to develop new preventative strategies, novel medicines, and more accurate risk assessment tools for OPMDs, ultimately improving patient outcomes and lowering the global burden of oral cancer. Food fortification and vitamin supplementation, particularly VitD, have emerged as potential strategies for reducing tumor burden and preventing malignant transformation. VitD has been recognized for its potential ability to increase cellular resistance to malignant transformations and influence the efficacy of cancer treatments through supplementation. Several research groups investigated the possible correlation between OPMDs and VitD deficiency, insufficiency, and sufficiency, as well as supplementation. The most commonly investigated OPMDs were oral lichen planus and leukoplakia. 

To summarize, current data is still not enough to draw final conclusions and offer recommendations. Nevertheless, the predominant data support the existence of a link between VitD insufficiency and the development or progression of OPMDs, considering the fact that VitD is essential for maintaining oral mucosal health. Hence, VitD deficiency may contribute to the etiology of several illnesses. While the link between VitD insufficiency and OPMDs is becoming better recognized, further studies are needed to establish a causal relationship and define the appropriate role of VitD supplementation in preventing or treating these lesions. The prevalence of VitD deficiency in people with OPMDs emphasizes the necessity of correcting this nutritional deficiency, aiming to decrease the incidence and enhance the prognosis of oral diseases. 

Encouraging outcomes from studies demonstrate reduced inflammation, improved immune response, and decreased tumor growth with VitD supplementation in various cancers. Additional clinical trials are being conducted to investigate the possible therapeutic benefits of VitD supplementation in people with OPMDs. However, further research is needed to define its specific role, optimal dosage, and evidence-based guidelines for OPMD management. VitD supplementation should be used alongside traditional measures and under healthcare professional supervision. Due to variations in study designs and the absence of standardized supplementation/treatment protocols among the analyzed clinical studies, there may be a variable risk of bias in the observed clinical studies.

It is important to note that the classification of OPMDs may continue to evolve as our understanding of these conditions improves. The WHO classification provides a framework for diagnosing and managing OPMDs and assessing the risk of malignant transformation [2]. Yet, regular monitoring and appropriate interventions are recommended for individuals diagnosed with OPMDs, to detect any signs of progression to oral cancer at an early stage. 

Although the discussed molecular mechanisms provide insight into the potential relationship between VitD and OPMDs, research in this area is still emerging, and many aspects remain to be fully understood. Further studies are needed not only to elucidate the precise molecular pathways involved but also to identify the best effective treatments for optimizing VitD status in people at risk of OPMDs.

Conclusively, using VitD-based therapies to prevent or cure OPMDs presents both challenges and opportunities. Among the future challenges will be confirming the role VitD plays in OPMDs, as well as determining an effective and safe supplementation schedule by randomized controlled trials (RCTs) evaluating VitD supplementation as an adjuvant therapy for OPMDs (Figure 4). Individual differences in VitD metabolism and the possibility of VitD toxicity must also be considered. Mechanistic research is also required to identify the particular cellular and molecular pathways by which VitD influences the development and progression of OPMDs. Such efforts will help to improve our understanding of the link between VitD and OPMDs, ultimately shaping evidence-based therapeutic guidelines.

## Figures and Tables

**Figure 1 ijms-24-15058-f001:**
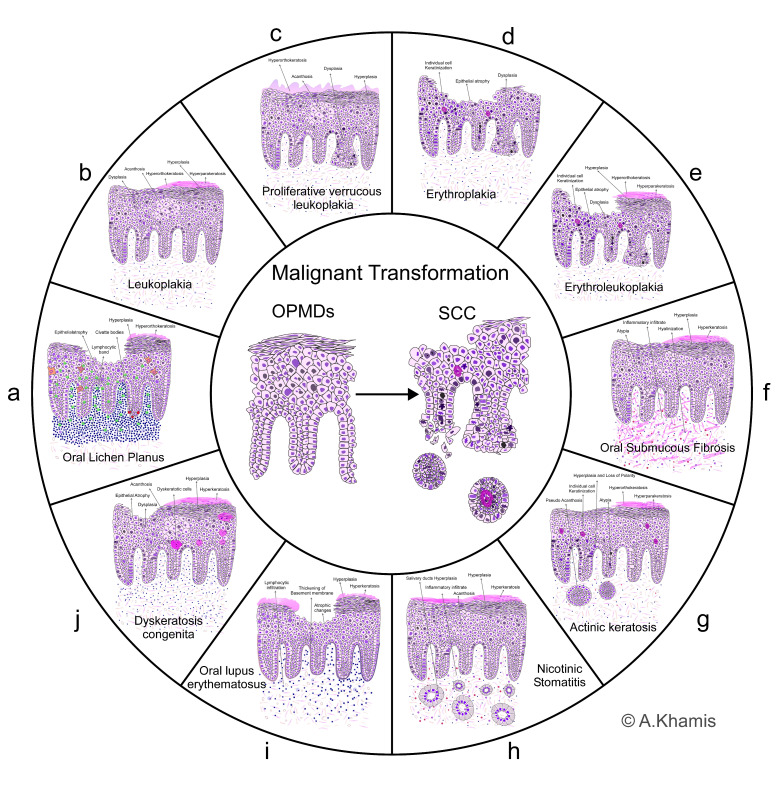
Overview of oral potentially malignant disorders. Histopathological illustration depicting various oral potentially malignant disorders and highlighting the potential sequelae of malignant transformation.

**Figure 2 ijms-24-15058-f002:**
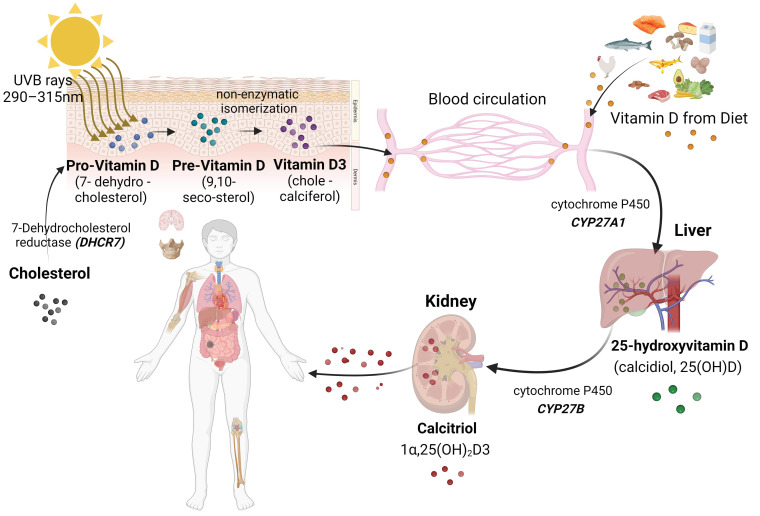
Vitamin D Metabolism. Schematic illustration representing the successive stages involved in VitD production and activation within the human body, as well as the metabolic pathways responsible for VitD utilization. Created with BioRender.com accessed on 30 September 2023.

**Figure 3 ijms-24-15058-f003:**
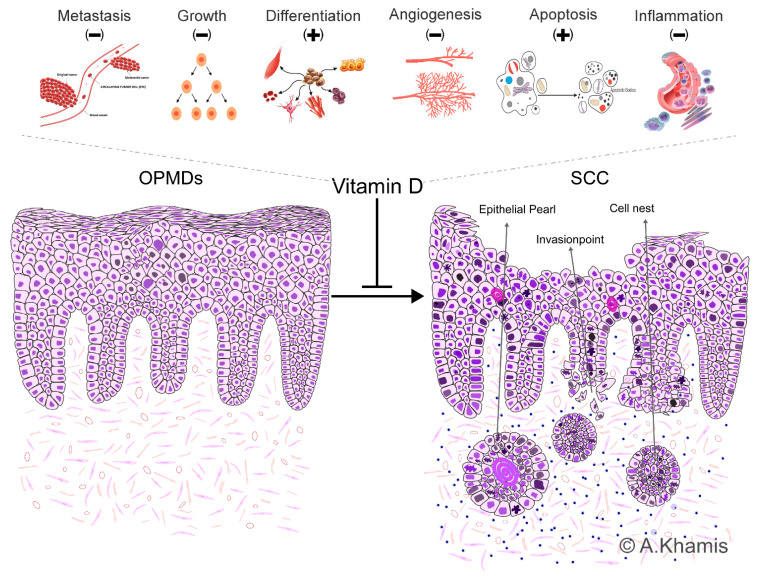
Sequela of Oral Potentially Malignant Disorders (OPMDs) and the Influence of Vitamin D on Lesion Progression. Illustration depicting the potential consequences of oral potentially malignant disorders (OPMDs) and highlighting the biological functions of VitD that influence the progression of oral lesions. “+” indicates induction/enhancement, “−” indicates inhibition/decrease by VitD.

**Figure 4 ijms-24-15058-f004:**
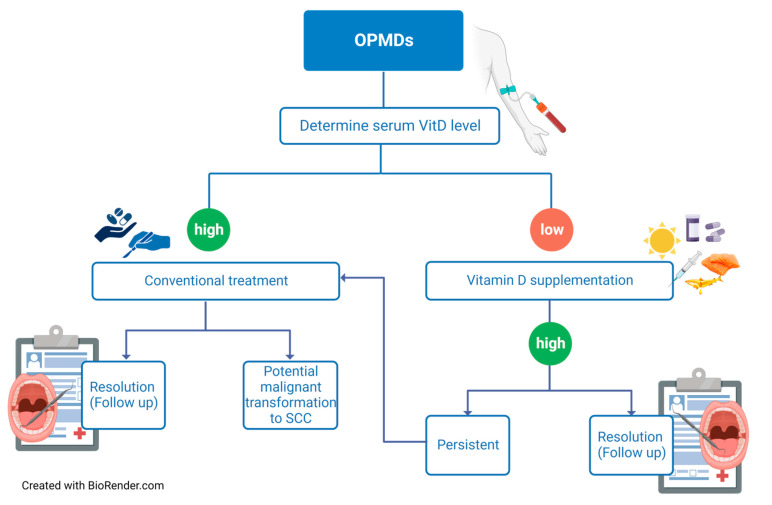
Flow Chart Depicting Vitamin D Supplementation Strategies for OPMD Treatment. Schematic representation illustrating various treatment modalities for OPMDs with respect to VitD supplementation. The treatments are categorized based on serum VitD levels following diagnosis. ‘Low’ indicates deficient or insufficient VitD levels, while ‘High’ signifies sufficient VitD status. Created with BioRender.com accessed on 30 September 2023.

**Table 1 ijms-24-15058-t001:** Evaluation criteria for leukoplakia [7].

Criteria	Leukoplakia
Color	Primarily white
Appearance/shape	Homogenous: uniform, well-defined borders Non-homogenous: speckled, nodular with diffuse borders
Scrappability	Non-scrapable Remains after tissue retraction or stretching
Exclusion criteria	Presence of (chronic) traumatic irritation Signs or symptoms of other red/white lesions
Persistent	Persists after elimination of apparent traumatic causes

**Table 2 ijms-24-15058-t002:** Clinical characteristics of different OPMDs.

OPMDs	Site	Clinical Presentation	Key Histopathologic Features	Etiology	Demographics
**Oral lichen planus**	-Bilateral post buccal mucosa -Tongue -Lips and gingiva -Floor of the mouth	-Reticular -Plaque -Atrophic -Erosive -Bullous	-Hyperkeratosis, acanthosis, or atrophy -Saw-tooth rete-pegs -Atypia and dysplasia	Multi-factorial: -Immunological -Environmental -Drug reaction Stress	Sex: Female > Men Age: >40
**Leukoplakia**	-Vermillion border -Buccal mucosa -Gingiva -Palate -Tongue -Floor of the mouth	-White patch or plaque -Cannot be scraped off	-Mild to severe dysplasia	-Smoking -Alcohol -Sunlight -Nutritional deficiency -Infectious	Sex: Female = Male Middle Age
**Proliferative verrucous leukoplakia**	Multifocal	-Early: Flat White lesion -Late: warty, papillary or verrucous	-Dysplasia -Invasion may also be seen	Unknown	Sex: Female > Male Age: >40
**Erythroplakia**	Any	-Red Mucosal lesion -Cannot be scraped off -Sharply demarcated	-Mild to severe dysplasia -Carcinoma in situ -Invasion	Unknown	A common precursor of OSCC
**Erythroleukoplakia**	Any	Mixed mucosal lesion	Mild to severe dysplasia	Unknown	A common precursor of OSCC
**Oral submucous fibrosis (OSF)**	-Buccal mucosa -Retro molar area -Soft palate	-Marble-like pallor mucosa -Leathery appearance	-Sub-epithelial fibrosis and hyalinization -Epithelial atypia	-Chronic use of betel quid -Spicy food	Sex: Female > Male Age: young adult
**Actinic keratosis (AK)**	-Vermilion border -Lower lip	-Rough, scaly patches	-Epithelial atrophy -Ortho/Para keratinization -Atypia -Individual cell keratinization -Pseudo acanthosis	Sunlight	Sex Male > Female Middle age White skin
**Nicotinic stomatitis**	-Palate	-Thickened white plaques -Mucosal nodularity -Erythema -Ulceration	-Hyperkeratosis -Acanthosis -Squamous metaplasia -Salivary gland ducts hyperplasia	Reverse smoking	India Caribbean Islands Latin America Sardinia
**Oral lupus erythematosus (OLE)**	-Buccal mucosa -Palate -Lips	-Central circular zone of atrophic mucosa -White striae on borders	-Atrophic to hyperkeratotic epithelium -Sub-epithelial lymphocytic infiltration. -Basal cell degeneration -Thickening of the basement membrane	-Subtype of lupus erythematosus	Sex: Female > Male Age: First diagnosis 20–40 years
**Dyskeratosis congenita (DKC)**	-Tongue -Buccal mucosa -Soft palate	-White patch -Papillary projections	-Dyskeratotic cells -Epithelial atrophy -Acanthosis -Dysplasia -Hyperplasia -Hyperkeratosis	-Genetic disorder Zinsser-Cole-Engman syndrome	Young age

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
