# Peer review of "The Magic Triangle in Oral Potentially Malignant Disorders: Vitamin D, Vitamin D Receptor, and Malignancy"

_ijms, 2023, doi:10.3390/ijms242015058_

Round 1
Reviewer 1 Report
The paper by Khamis et al. is an interesting work about Vitamin D, its receptor, and OPMD.
In line or at least arrest 223 "their progression by employing effective preventive strategies." is a misleading sentence for potentially malignant disorders. I suggest the authors to rephrase the section.
I suggest the authors to add images or tables to make the paper easier to read without being verbose.
The manuscript is a catchy review about the influence of vitamin D and Vitamin D on Oral Potentially malignant disorders. The topic has been previously discussed for some of the OPMD; but the literature is missing of a systematic analysis of all of the OPMD. The text is easy to read and the conclusion are in line with the topics described since the paper is a non clinical research.
The references are not formatted following the guidelines of the journal
Author Response
Please find the response in the attachment

Reviewer 2 Report
The authors made an attempt to review a scientific evidence linking Vitamin D and Oral Potentially Malignant Disorders using a simplified scoping review approach. Although the primary concept seems undoubtedly intriguing considering hypothetical association between vitamin-related metabolic processes and neoplasm transformation, numerous concerns must be clarified before further proceeding.
Major comments:
1. Speculations and hypothetical assumptions overtake the scientific bases of primary concept. Overoptimistic approach applied by authors may not reflect a current research evidence.
2. Lack of systematic review and similarly, lack of critical evaluation of existing evidence.
3. No standardized assessment of the quality of reviewed studies utilising standardised criteria.
Minor comments.
1. Figure 1 does not provide any essential added value towards the presented information and hypothesis discussed.
2. Missing authors' contributions.
Overall, the manuscript by Khamis et al. requires substantial amendments comply with scientifically sound research paper.
English language is used meets fully scientific criteria.
Author Response
Please find the response in the attachment.

Reviewer 3 Report
Dear Authors,
It was a pleasure to read the manuscript. The article is very well written, structured in a clear manner, with latest information from the scientific literature.
I appreciate the presentation of the OPMD in the first part of the article. Every OPMD was presented for a better understanding.
Regarding the sections of the manuscript where the Vitamin D and it s implication in prevention of the head and neck cancer, i appreciate the debate and the informations presented.
Please revise the section with author contributions.
The manuscript language is ok, without needing english corrections.
From my point of view, the article can be published!
Author Response

(The authors gave the same response as above.)

Round 2
Reviewer 2 Report
The authors provided sufficient explanations followed first round of review. Essentially, it is still crucial to highlight firmly in 'limitations' section that a robust systematic review is required to validate the authors' primary hypotheses.